# Prognostic Utility of Platelet–Lymphocyte Ratio, Neutrophil–Lymphocyte Ratio and Monocyte–Lymphocyte Ratio in Head and Neck Cancers: A Detailed PRISMA Compliant Systematic Review and Meta-Analysis

**DOI:** 10.3390/cancers13164166

**Published:** 2021-08-19

**Authors:** Chellan Kumarasamy, Vaibhav Tiwary, Krishnan Sunil, Deepa Suresh, Sameep Shetty, Gothandam Kodiveri Muthukaliannan, Siddhartha Baxi, Rama Jayaraj

**Affiliations:** 1School of Health and Medical Sciences, Curtin University, Perth, WA 6102, Australia; chellank54@gmail.com; 2School of Biosciences and Technology, Vellore Institute of Technology (VIT), Vellore 632014, India; vaibhavtiwary1397@gmail.com (V.T.); gothandam@gmail.com (G.K.M.); 3Department of Radiation Oncology, Mayo Clinic Florida, Jacksonville, FL 32224, USA; Krishnan.Sunil@mayo.edu; 4Division of Endocrinology, Department of Internal Medicine, Mayo Clinic Florida, Jacksonville, FL 32224, USA; Deepa.Suresh@mayo.edu; 5Department of Oral and Maxillofacial Surgery, Manipal College of Dental Sciences, Mangalore, Manipal Academy of Higher Education, A Constituent of MAHE, Manipal 576104, India; sameep.shetty@manipal.edu; 6Radiation Oncology, Genesiscare Gold Coast, John Flynn Hospital, 42 Inland Drive, Tugun, QLD 4224, Australia; Siddhartha.baxi@genesiscare.com; 7Northern Territory Institute of Research and Training, Darwin, NT 0909, Australia

**Keywords:** PLR, NLR, MLR, HNC, prognosis, meta-analysis, systematic review

## Abstract

**Simple Summary:**

Inflammation plays a major role in cancer development and progression and has the potential to be used as a prognostic marker in cancer. Previous studies have attempted to evaluate PLR, NLR and MLR as indicators of inflammation/prognostic markers in cancer, but there is no common consensus on its application in clinical practice. The aim of this systematic review and meta-analysis (a) assess the prognostic efficacy of all three prognostic markers in comparison to each other and, (b) investigate the prognostic potential of these three markers in HNC. The study followed PRISMA guidelines, with literature being collated from multiple bibliographic databases. Preliminary and secondary screening were carried out using stringent inclusion/exclusion criteria.

**Abstract:**

Inflammation plays a major role in cancer development and progression and has the potential to be used as a prognostic marker in cancer. Previous studies have attempted to evaluate Platelet-to-lymphocyte ratio (PLR), neutrophil–lymphocyte ratio (NLR) or monocyte–lymphocyte ratio (MLR) as indicators of inflammation/prognostic markers in cancer, but there is no common consensus on their application in clinical practice. The aim of this systematic review and meta-analysis is to (a) assess the prognostic efficacy of all three prognostic markers in comparison to each other and (b) investigate the prognostic potential of these three markers in HNC. The study followed PRISMA guidelines, with the literature being collated from multiple bibliographic databases. Preliminary and secondary screening were carried out using stringent inclusion/exclusion criteria. Meta-analysis was carried out on selected studies using CMA software and HR as the pooled effect size metric. A total of 49 studies were included in the study. The pooled HR values of PLR, NLR and MLR indicated that they were significantly correlated with poorer OS. The pooled effect estimates for PLR, NLR and MLR were 1.461 (95% CI 1.329–1.674), 1.639 (95% CI 1.429–1.880) and 1.002 (95% CI 0.720–1.396), respectively. Significant between-study heterogeneity was observed in the meta-analysis of all three. The results of this study suggest that PLR, NLR and MLR ratios can be powerful prognostic markers in head and neck cancers that can guide treatment. Further evidence from large-scale clinical studies on patient cohorts are required before they can be incorporated as a part of the clinical method. PROSPERO Registration ID: CRD42019121008

## 1. Introduction

It is an established fact that cancer pathophysiology relies heavily on the manipulation of the immune system involved in cancer cell growth, proliferation and tumorigenesis [1]. Immunological involvement, or, more specifically, inflammation, has been defined as one of the hallmarks of cancer and plays a role in malignancy, angiogenesis and genomic instability [2]. Studies have shown that the presence of inflammation and inflammatory markers in cancer is associated with a change in prognosis [3]. Therefore, an assessment of the magnitude of inflammation in cancer patients could be used to determine disease prognosis. The magnitude of inflammation could be indirectly explored via the measurement of malnutrition and systemic inflammation-based indicators such as neutrophil–lymphocyte ratio (NLR), platelet–lymphocyte ratio (PLR) and monocyte–lymphocyte ratio (MLR). NLR can act as a proxy measurement of the degree of inflammation in cancers, as inflammation leads to systemic alterations in the levels of peripheral blood leukocytes [4,5]. Similarly, platelets release pro-inflammatory mediators, such as cytokines and chemokines, which intensify the inflammatory microenvironment in tumors, resulting in PLR being another viable measure of inflammation [6,7]. In addition, monocytes also have a key role in inflammation, as seen in their presence in atherosclerosis [8]. Individually, or in combination, these measures could be used as prognostic markers in cancer. The dense immune cell influx may explain the robust immune response in HPV-driven head and neck cancers and its potential to eliminate some of the tumor cells. An assessment of the prognosis of patients with cancer is significant, since a precise prediction of prognosis will permit appropriate treatment, improving patient outcomes [9]. The systemic inflammatory response elicited by the immune cells seen in cancer is interconnected with the nutritional depletion seen in cancer patients and affects the prognosis and course of the disease independently of tumor stage [10]. The host immune system is involved in cancer initiation, progression and metastasis. In addition, the immune–inflammatory retort to the toxic chemoradiation therapy adds to its clinical benefits. Lymphocytes represent one third of the total white blood cells, play a significant role in cellular immunity and may stimulate the clearance of malignant cells [11]. Antagonizing the action of lymphocytes, the monocytes enable tumor progression by promoting angiogenesis and immunosuppression [12]. An assessment of the MLR at different stages pretreatment or post-treatment can help in anticipating the prognosis of the disease and improving treatment decisions [13].

The relative ease of assessment, swift detection and minimal cost using NLR, PLR and MLR offers a lucrative option to gauge cancer prognosis and guide treatment. Furthermore, the need for only peripheral blood samples with little to no patient discomfort or pain increases the patient compliance [14]. Current methods for cancer prognosis in patients involve the use of molecular markers (such as BRCA1 in breast cancer and EGFR in NSCLC) [15,16], which require complex and expensive assays for measurement and quantification (immunohistochemistry, q-RT PCR) [17,18], while also generating a greater degree of patient discomfort (biopsy), with an additional risk of seeding of tumor cells. Interpretation of tumor markers with complex assays does offer vital information on cancer prognosis and tumor-specific immunotherapy but they are not ideal for the constant and real-time monitoring of prognosis, as they are expensive, cumbersome and time-consuming. Therefore, significant research interest has been directed towards the use of PLR, NLR and MLR as additional or supplementary biomarkers for cancer prognosis.

Clinical studies have further explored the use of PLR, NLR and MLR as pre-operative prognostic biomarkers in addition to assessing their utility as diagnostic cancer biomarkers [19,20]. Further, the vast literature on these serum biomarkers has provided an impetus to conduct systematic review and meta-analysis studies [21,22]. However, a knowledge gap that exists, despite the presence of previous reviews, is that no systematic review or meta-analysis study published to date has investigated either (a) the prognostic efficacy of all three prognostic markers, PLR, NLR and MLR, in comparison to each other, or (b) the comparative utility of these markers with regard to HNC cancers. Previous systematic review and meta-analysis studies have either only focused on a single cancer type, with no study focusing on assessing the effectiveness of all three proposed biomarkers in HNC, and/or comparing and contrasting all three biomarkers against each other. While Mellor et al. recently published a systematic review and meta-analysis study, including all cancer types, the study only focused on NLR alone as the prognostic marker of choice [23]. Similarly, Zhu et al. focused on multiple inflammatory markers, with emphasis on PLR and NLR, but limited their study to ovarian cancer [24], while another study by Zhang et al. limited their analysis to only colorectal cancer [21].

This comprehensive systematic review and meta-analysis study seeks to amend the aforementioned knowledge gap and attempts to provide a better understanding of the utility of PLR, NLR and MLR as prognostic markers in cancer. In addition, it seeks to highlight the comparative efficacy of PLR, NLR and MLR in each studied cancer type.

## 2. Methods

### 2.1. Search Strategy

This systematic review and meta-analysis study was conducted based on the standard review guidelines and methodology as detailed in the Preferred Reporting Items for Systematic Review and Meta-Analyses Protocol (PRISMA-P) guidelines [25,26]. The PRISMA compliance has been delineated in the PRISMA checklist table provided in Appendix A. The search strategy was expansive and exhaustive, with multiple bibliographic databases being searched. These databases included EMBASE, MEDLINE, Science Direct, Scopus and Web of Science. These databases were used to scope out relevant studies published within a ten-year span of time, i.e., from July 1999 to July 2019. This time constraint was placed to the search strategy in order to keep the results of the search relevant to the current developments in cancer prognosis and treatment. The search strategy also allowed for scoping of the reference lists of review articles and other publications for additional relevant studies for inclusion in the review. The bibliographic search was conducted via the construction of a logic grid and subsequent identification of specific ‘keywords’. These ‘keywords’ were then used to construct ‘search strings’, which allowed for a thorough and robust search of the bibliographic databases.

Keywords: Platelet-to-Lymphocyte Ratio (PLR), Neutrophil-to-Lymphocyte Ratio (NLR), Monocyte-to-Lymphocyte Ratio (MLR), Biomarker, Head and Neck Cancer, Prognosis, Survival (Overall Survival, Disease Free Survival, Disease Specific Survival), Patient Study, Cohort, Blood, Systematic-review, Meta-analysis.

The search was conducted individually by two independent reviewers (CK and RJ), in order to eliminate the likelihood of selection bias occurring. Primary article and study screening was based on the pertinence of the title and abstract of each publication to the topic of the systematic review and meta-analysis being undertaken. The screening was conducted simultaneously alongside the initial search, under the discretion of the two reviewers (CK and RJ). Additional studies from the reference lists of reviews and other included publications were screened for and included after primary screening. Any disputes and differences in opinion arising during initial screening were settled through the inclusion of the third reviewer (VT).

### 2.2. Study Selection

After the primary screening procedure, the full texts of articles were subject to selection based on specific, predefined inclusion and exclusion criteria (secondary screening). The inclusion and exclusion criteria were based on previous similar systematic review and meta-analysis studies, adapted to the parameters of this study.

### 2.3. Inclusion Criteria

The studies must discuss the survival outcome of HNC cancer patients based on PLR, NLR and MLR levels.The survival outcome must be presented in the form of HR (hazard ratios) and 95% CI (confidence intervals).The survival outcome must be presented in the form of Kaplan–Meier curves, along with patient cohort information, for each treatment arm represented in the KM Curves. (This is required only if the HR and 95% CI values have not been presented in the manuscript, as the above information is required to extract approximated HR values.)

### 2.4. Exclusion Criteria

Conference abstracts, reviews, letters to the editor and other non-clinical literature will not be considered for either systematic review or meta-analysis.Included studies must be clinical studies or involve patient samples. (In vitro, in silico and animal studies will be excluded.)Unpublished or non-peer-reviewed literature will be excluded.Studies that do not focus on survival outcomes and prognosis aspects of PLR, NLR and MLR in cancer patients will not be considered.If the sample size of each individual study is of low power (sample size < 10), they will be excluded.

No limitations were placed on the types of patients involved, their clinicopathological characteristics or their demographic characteristics. No restrictions were placed based on age, sex, ethnicity, location, follow-up period, duration of treatment or method of treatment.

### 2.5. Data Extraction and Recording

After the secondary screening procedure, based on the predefined inclusion and exclusion criteria, the full-text formats of the selected studies were collated and subjected to a data extraction process. Data extraction followed a top-down approach, with the selected full-text studies being examined for relevant patient and study data individually by three reviewers, a process designed to generate redundancy, while reducing individual error. A standardized data extraction form, containing all the required data items, was prepared using Microsoft Excel and utilized by the reviewers to extract the data. After individual data extraction was performed by all reviewers, duplicated information/studies were removed. The resulting dataset of study information was collated into a single database for further analysis. The data items extracted from full-text versions of individual studies included:Author names;Year of publication;Marker studied (PLR, NLR or MLR);Size of patient cohort;Diagnostic methods;Follow-up period;Gender split of cohort;TNM staging split of cohort;Survival endpoint of each study (overall survival, disease-free survival, disease-specific survival);General features of each study (will be presented as short qualitative opinions/observations of reviewers, for each study being included).

### 2.6. Quality Assessment

The quality assessment of the studies was based on the standard Newcastle–Ottawa Scale (NOS) for the quality assessment of nonrandomized studies in meta-analysis [27]. This scale presents a ‘star system’, which assigns each of the 3 broad parameters of the study (1–4 stars range) in increasing order of quality. The 3 broad perspectives being assessed include the selection of study groups in each individual study, the comparability of groups and the ascertainment of exposure/outcome of interest for case–control/cohort studies [28,29,30,31,32].

### 2.7. Meta-Analysis

The meta-analysis was conducted using the Comprehensive Meta-Analysis (CMA) software (version 3.3.070; Biostat, Englewood, NJ, USA) [33]. The hazard ratio was chosen as the appropriate effect size metric for this study, and overall survival (OS) was selected as the survival endpoint for the meta-analysis. The HR indicates the probability of survival of the patient cohort in each included study and was pooled across all included studies to determine the likelihood of overall survival (OS) of patients, across all studies. The pooled results were represented visually using forest plots. Meta-analyses on PLR, NLR and MLR were performed independently, to determine the prognostic effectiveness of each, as a cancer marker. Statistical significance (*p*-values) of each of the three aforementioned prognostic markers was also calculated. The meta-analysis was conducted based on the random-effects model to account for inherent heterogeneity between each of the included studies [34].

### 2.8. Assessment of Heterogeneity

Assessment of heterogeneity was conducted using 3 parameters in order to increase the robustness of analysis [35,36,37,38,39]. The Higgins I^2^ statistic was used as the primary method to determine heterogeneity, as it has a high power of detection of heterogeneity [40]. However, as I^2^ is not an infallible metric of heterogeneity, and can provide biased results in small meta-analyses [41], we also used the Cochran’ Q and Tau^2^ parameters to assess heterogeneity alongside the I^2^ statistic [42,43].

### 2.9. Subgroup Analysis

The major meta-analysis subgroups selected for this study were cohorts of studies that assessed PLR, NLR and MLR. As sufficient data were not available, only the survival endpoint of OS was assessed. Additionally, no subgroup analysis was carried out based on the demographic or clinicopathological characteristics, due to a lack of sufficient high-quality studies with comparable data, which would lead to a lack of power in the subsequent statistical analysis.

### 2.10. Publication Bias

Publication bias assessment was conducted as per PRISMA guidelines. The Egger’s graphical bias indicator test was used to construct a funnel plot [44]. The funnel plot symmetry along the regression line was used to assess the existence of publication bias, wherein the symmetry of the funnel plot inversely correlated to the degree of publication bias in the meta-analysis study. Adjustment for small and missing studies was carried out via imputation of possible small studies using the Orwin’s Fail-Safe N test [45]. The Begg and Mazumdar’s Rank Correlation test was used to check correlation between ranks of effect sizes and variances [46], wherein a positive result indicated accurate publication bias assessment.

## 3. Results

The search strategy yielded a total of 28,716 studies across all databases. The majority of these studies were screened out by reviewers during the initial primary screening process, due to a lack of relevance to the topic of review. After primary screening, full texts of 120 articles were obtained, which were then further assessed for duplicates (which were removed) and adherence to pre-defined inclusion and exclusion criteria. This secondary screening process eliminated 71 studies, which left 49 studies suitable for inclusion in the systematic review and meta-analysis. However, due to a lack of sufficient quality data in all screened studies, only 34 publications that had the appropriate effect size data (HR and 95% CI values for OS) were included in the final meta-analysis. The data were subsequently extracted from the selected studies as per the defined data extraction procedure and used to construct a systematic review table and perform the meta-analysis. The entire process was monitored by the second and third reviewer (RJ and VT) at all stages. The selection process is delineated in Figure 1.

Study Characteristics: The 49 studies included in this systematic review and meta-analysis study were conducted across nine countries across the globe, as described in Table 1. The largest number of published studies were from China (*n* = 28). Other countries that featured in multiple publications on this topic included Japan (*n* = 9), the United Kingdom (*n* = 3), Taiwan (*n* = 3) and the Republic of Korea (*n* = 2), while Italy (*n* = 1), Switzerland (*n* = 1), Thailand (*n* = 1) and Turkey (*n* = 1) each only featured in a single publication.

Most of these studies were large-scale retrospective studies, representing a total pooled patient cohort of 20,739 patients. The collated data across these 49 studies suggest that nearly all studies in this field of PLR-, NLR- and MLR-based cancer prognostics have focused on head and neck cancers (HNC), with the main anatomical locations described being the oral cavity, nasopharyngeal, oropharyngeal, hypopharyngeal, laryngealnasopharyngeal, laryngeal and tongue regions. Further observation of the demographic data also suggests that most studies included in this review involved a higher percentage of men, in comparison to women (*n* = 45), with only a few studies reporting a higher percentage of women in their studies (*n* = 4). Not all studies assessed all three prognostic indicators of PLR, NLR and MLR. A large proportion of the studies assessed PLR and NLR in conjunction (*n* = 24), with a smaller proportion assessing all three simultaneously (*n* = 13). A few studies were also noted to have assessed each of the prognostic markers individually, with PLR (*n* = 6) and MLR (*n* = 3) both being assessed equally, while no study assessed NLR individually (*n* = 0). While most studies did not present any information regarding underlying risk factors (*n* = 39), the studies that did provide this information highlighted smoking (*n* = 10) and drinking (*n* = 8) as the primary risk factors, while one study also detailed additional risk factors such as chronic HPV infection, CVD, diabetes and family history.

## 4. Meta-Analysis

As described in the methodology and protocol of the study, a meta-analysis of pooled HR and 95% CI values was carried out under three subgroups, for the survival endpoint of OS. The subgroups analyzed were based on the three potential prognostic markers PLR, NLR and MLR. Each individual cohort of each study focusing uniquely on each of the three aforementioned prognostic markers was assessed as an independent study, contributing to the respective subgroup being analyzed. The pooled results were graphically represented in the form of forest plots.

### 4.1. Meta-Analysis PLR Subgroup

A total of 25 cohorts of studies were pooled for determining the prognostic impact of a change (increase) in PLR levels upon the overall survival of the patient cohort (Figure 2). Twenty-three studies showed a positive correlation between increased PLR levels and a worse disease outcome (poorer prognosis), out of which 18 studies showed a significant level of said correlation (*p* < 0.05). Interestingly, two other additional studies contradicted the rest, suggesting an inverse correlation between PLR levels and patient survival, thereby suggesting a better prognosis for OS. The pooled effect estimate (HR) was found to be statistically significant, at a value of 1.461 (95% CI 1.329–1.674; *p* = 0.0001). Assessment of heterogeneity suggests that there is substantial heterogeneity between the studies included in the subgroup meta-analysis for PLR (I^2^ = 80.320; Tau^2^ = 0.050; Cochran’s *Q* = 121.949).

### 4.2. Meta-Analysis NLR Subgroup

A total of 27 cohorts of studies were pooled for determining the prognostic impact of a change (increase) in NLR levels upon the overall survival of the patient cohort (Figure 3). The majority of the studies (*n* = 23) showed a positive correlation between increased NLR levels and a worse disease outcome (poorer prognosis), out of which 20 studies showed a significant level of said correlation (*p* < 0.05). Interestingly, only one other study contradicted the majority, suggesting an inverse correlation between NLR levels and patient survival, indicating a better prognosis for OS. The pooled effect estimate (HR) was found to be statistically significant, at a value of 1.639 (95% CI 1.429–1.880; *p* = 0.001). Assessment of heterogeneity suggests that there is substantial heterogeneity between the studies included in the subgroup meta-analysis for NLR (I^2^ = 82.152; Tau^2^ = 0.085; Cochran’s *Q* = 145.674).

### 4.3. Meta-Analysis MLR Subgroup

Fewer of the included studies focused on MLR as a prognostic marker, in comparison to PLR and NLR, with a total of only 12 studies being available for inclusion in the assessment of the prognostic efficacy of MLR (Figure 4). In contrast to PLR and NLR, in MLR, the majority of the studies (*n* = 7) showed a negative correlation between increased MLR levels and a worse disease outcome (poorer prognosis), which indicates an increase in patient survival (only 1 study out of the aforementioned 7 had a non-significant effect size, *p* < 0.05). However, the rest of the included studies (*n* = 5) instead showed a contradictory result, where all five studies indicated a statistically significant effect of MLR levels of poorer patient prognosis (*p* < 0.05). Overall, the pooled effect estimate (HR) was found to not be statistically significant, with a pooled HR value of 1.002 (95% CI 0.720–1.396; *p* = 0.989). Assessment of heterogeneity suggests that there is substantial heterogeneity between the studies included in the subgroup meta-analysis for PLR (I^2^ =93.902; Tau^2^ = 0.292; Cochran’s *Q* = 180.381).

### 4.4. Publication Bias

The Eggers’ graphical test was used to assess possible publication bias. The funnel plot was constructed using the same software used to conduct the meta-analysis, CMA (Ver 3.3.070; Biostat, Englewood, NJ, USA). The funnel plots are visualized in Figures (X), (Y) and (Z) for the subgroups, PLR, NLR and MLR, respectively. Orwin’s fail-safe N test was applied the publication bias assessment for the PLR and NLR subgroups, for the imputation of multiple missing studies. The funnel plot indicates the presence of publication bias in the PLR and NLR subgroups, which is observed in the significant skew in the distribution of studies along the line of mean effect. However, no significant publication bias was observed in the MLR subgroup.

### 4.5. Quality Assessment

The Newcastle–Ottawa scale (NOS) was used to assess the quality of all included studies. All included studies were found to have quality greater than 2 stars on the assessment scale, which was deemed to be satisfactory for the inclusion of said studies in the meta-analysis. Regardless, the main requirement of inclusion in the study was the availability of good-quality, extractable statistical data from each individual study.

## 5. Discussion

The current systematic review and meta-analysis were conducted to scrutinize and explore the prognostic potential of PLR, NLR and MLR ratios as prognostic markers in cancer. The use of these prognostic markers has progressed due to the existing established evidence that inflammation can drive cancer growth and progression [3]. In particular, the ease of accessibility, low cost and patient comfort associated with this analysis provide significant benefits if PLR, NLR and MLR are established as clinically reliable cancer prognosticators. Earlier patient cohort studies have investigated the possibility of the use of PLR, NLR and MLR as prognostic markers in cancer [21,22]. However, no comprehensive systematic review or meta-analysis currently exists detailing the comparative efficacy between the three prognostic markers.

A previously published meta-analysis study focused on NLR as a cancer prognostic marker in solid tumors, but did not explore or investigate MLR and PLR as potential prognostic markers in head and neck cancers [23]. Furthermore, while other meta-analysis studies have discussed the efficacy of PLR, NLR and MLR as inflammatory indicators in other chronic conditions, such as infectious diseases and psychosis, their effect in cancer still requires further exploration. While this study looked only at these parameters as predictors of overall outcomes, there remains the possibility that dynamic changes in these parameters during treatment may predict response to treatment. This is especially useful in a tumor such as HNC, especially human papilloma virus (HPV)-negative HNC, because there is currently no universally approved non-invasive and affordable biomarker that serves as a surrogate for tumor burden or treatment response. For HPV-positive cancers, serum levels of HPV DNA show promise as reliable surrogates for treatment response and tumor burden.

This systematic review and meta-analysis study that assessed 49 studies, across nine countries and involving 20,729 patients, investigated the effectiveness of peripheral blood PLR, NLR and MLR ratios in head and neck cancer prognosis. All three of the aforementioned prognostic markers were assessed separately during the meta-analysis as three individual subgroups. It was observed that, overall, more studies had evaluated the prognostic potential of NLR and PLR on patient survival, compared to MLR. The systematic review and meta-analysis study by Mellor et al. serves to validate the findings of this study with regard to PLR as a prognostic marker. While Mellor et al.’s study had a much smaller sample size of included studies (*n* = 5), for assessing patient OS when compared to this study (*n* = 25), the results obtained corroborate and lay a foundation for the findings garnered by Mellor et al., where an increase in PLR levels was found to be significantly associated with poor survival [23]. Furthermore, in the meta-analysis study conducted by Zhu et al., they assessed the effectiveness of PLR and NLR as prognostic markers in ovarian cancer [24]. Despite the difference in the types of cancer being investigated, with the current study assessing the prognostic potential of PLR and NLR in HNC and Zhu et al.’s study investigating ovarian cancer, the results of these twin studies indicated poor OS in cancer patients with higher PLR and NLR levels. A study similar to Zhu et al.’s was conducted by Zhang et al., wherein they focused on highlighting the prognostic potential of PLR and NLR in colorectal cancer. Zhang et al.’s study also reached a similar conclusion, with elevated NLR levels being indicative of poor overall survival in their study [21]. Previous studies, such as the systematic review and meta-analysis by Tham et al., indicate that MLR, unlike PLR and NLR, may have a positive prognostic effect, with elevated MLR levels being correlated with an improved OS [93]. The results of another study by Kano et al. also reflect this statement [88]. However, it is important to note that Tham et al.’s study had a small sample size, pooling five studies for OS, while Kano et al.’s study was a singular retrospective study. Interestingly, the studies used by Tham et al. overlap with the seven studies in our meta-analysis, where we observed a similar pattern of MLR leading to an improved OS. However, when considering the larger sample size of our meta-analysis for MLR, with a near equal number of studies (*n* = 5) showing a statistically significant negative effect of increased MLR levels on patient survival, MLR’s overall effect on OS remains inconclusive.

This study does have a few limitations that warrant being highlighted. The limited quantity of homogeneous, high-quality literature published in this field impeded detailed subgroup analysis based on clinicopathological and demographic criteria. Additionally, not all studies reported the HR and 95% CI values in a numerical form, which required HR and 95% CI values to be extracted from Kaplan–Meier curves for OS presented in these publications. As extracting numerical data from graphical representations involves estimation, there may be some degree of error introduced into the study, which must be considered when applying the results of this study in the clinical sphere.

Overall, this systematic review and meta-analysis study is validated by previous results seen in other studies in the published literature, while simultaneously bringing into question the results of other studies, by building upon them with a much broader scope of approach and a larger pool of literature incorporated into the meta-analysis. Therefore, the results presented in this paper provide evidence to suggest that PLR and NLR ratios may be capable of being used clinically, as prognostic markers in HNCs, and could, in conjunction with the traditional prognostic indicators of cancer, provide a more robust clinical analysis of patient survival and prognosis in HNC. With regard to MLR, however, there appears to be some contention on its effect on survival in HNC, indicating that further research is needed before a conclusive statement on its clinical utility as a prognostic marker in HNC is presented.

## 6. Conclusions

The study indicates that while PLR and NLR ratios have evidenced potential as prognostic markers for clinical use, particularly in HNC, MLR cannot be currently recommended for clinical use as a prognostic marker. Furthermore, we would like to highlight that the results presented here were obtained from the pooling of multiple individual studies, which each had their own study design, parameters and analysis methods. Therefore, despite the results presented here and previous studies validating these results, the inherent heterogeneity between individual studies being pooled requires that further large-scale clinical studies with a large sample size and a homogeneous approach are conducted before bringing these prognostic markers into clinical practice. Therefore, large-scale, longitudinal patient studies focusing on PLR and NLR as prognostic markers are necessary before they can be incorporated into standard practice as complementary biomarkers to currently existing prognostic markers in cancer. Until then, the results of this systematic review and meta-analysis serve to aid in both clinical decision-making and ongoing and future research in this field.

## Figures and Tables

**Figure 1 cancers-13-04166-f001:**
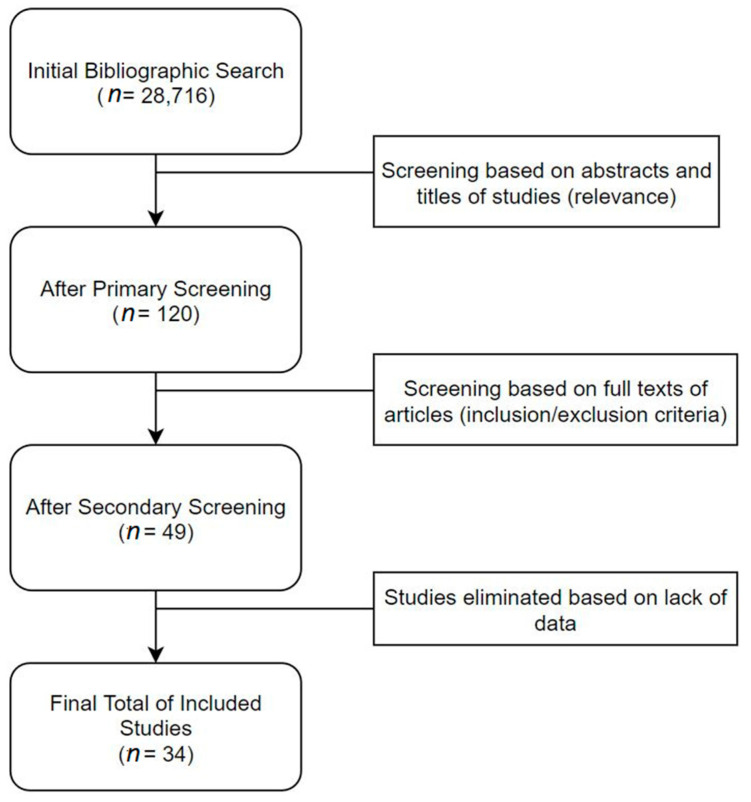
Search Strategy.

**Figure 2 cancers-13-04166-f002:**
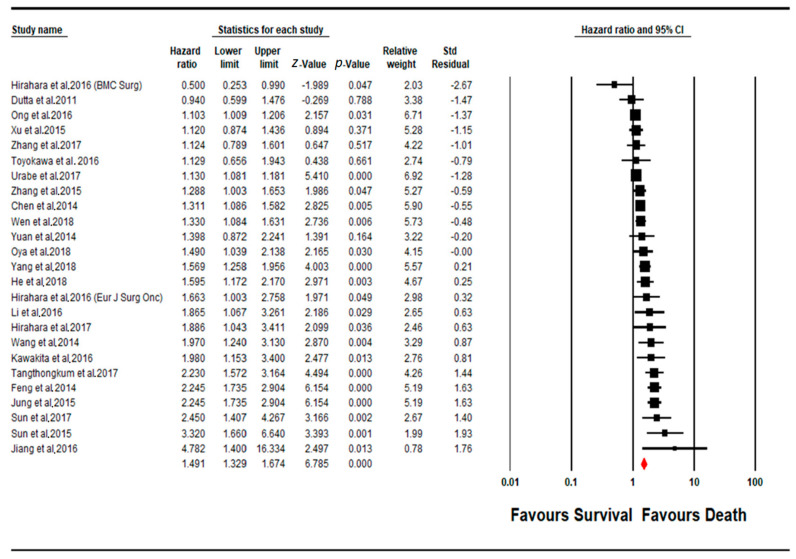
Forest plot of PLR subgroup.

**Figure 3 cancers-13-04166-f003:**
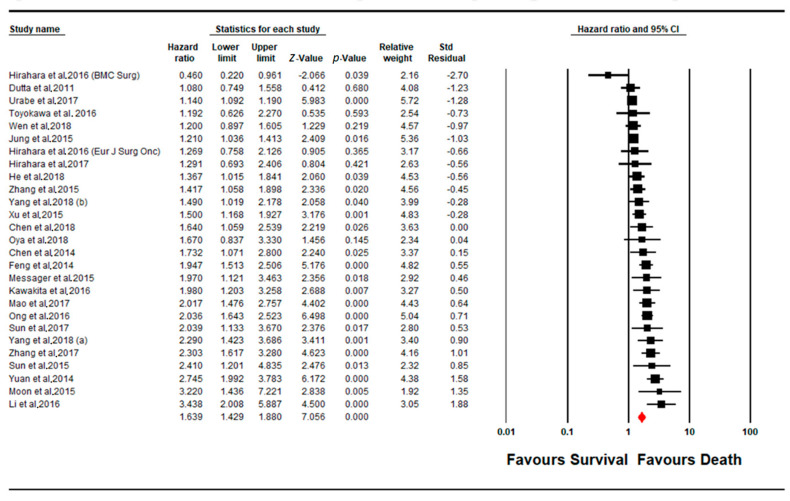
Forest plot of NLR subgroup.

**Figure 4 cancers-13-04166-f004:**
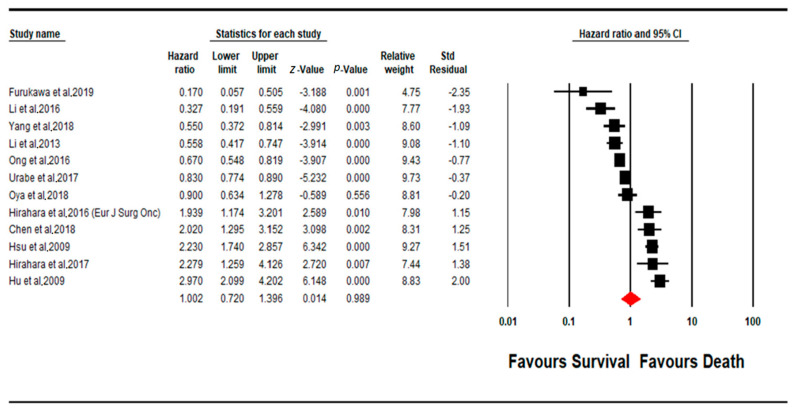
Forest plot of MLR subgroup.

**Table 1 cancers-13-04166-t001:** Study characteristics of included studies for systematic review.

Author Name	Year of Publication	Prognostic Parameter (PLR/NLR/MLR)	Cohort Size	Anatomic Location of Cancer	Country of Study	Type of Study	Gender	Stage of Cancer	Metastasis	Risk Factors	Age
Wen et al. [47]	2018	PLR NLR	723	Esophageal (65%) Gastric (35%)	UK	retrospective study	male (75.5%) female (24.5%)	T0 (4.6%) T1 (14.4%) T2 (21.0%) T3 (51.9%) T4 (8.2%)	M1 2.5%	NA	66.1 ± 10.5
Kawakita et al. [48]	2016	PLR NLR	140	Parotid gland (78%) Submandibular gland (20%) Others (2%)	Japan	retrospective study	male (86%) female (14%)	T1 (9%) T2 (26%) T3 (20%) 4a (44%) 4b (1%)	M0 (93%) M1 (7%)	NA	64 (26–84)
Chen et al. [49]	2014	NLR PLR	211	Nasopharynx (100%)	China	retrospective study	male (85.8%) female (14.2%)	NA	NA	NA	46 (14–72)
Zhang et al. [50]	2015	NLR PLR	468	Esophagus (100%)	China	retrospective study	male (80.3%) female (19.7%)	stage I (9.8 %) stage II (42.6 %) stage III (47.6 %)	NA	Smoking Alcohol consumption	59.5 ± 9.0
Jiang et al. [51]	2017	PLR NLR LMR	78	Thyroid (100%)	China	retrospective study	male (43.6%) female (56.4%)	stage I (23.1%) stage II (20.5%) stage III (20.5%) stage IV (35.9%)	M1 (2.6%)	NA	47.3 ± 13.8
Li et al. [52]	2016	PLR NLR LMR	388	Nasopharynx (100%)	China	retrospective study	NA	NA	M0 (100%)	NA	
Feng et al. [53]	2014	NLR PLR	483	Esophagus (100%)	China	retrospective study	male (85.1%) female (14.9%)	T1 (18.0%) T2 (16.6%) T3 (54.9%) T4 (10.5%)	M0 (56.7%) M1 (43.3%)	NA	59.1 ± 8.0
Jiang et al. [54]	2015	PLR	1261	Nasopharynx (100%)	China	retrospective study	male (72.9%) female (27.1%)	Clinical stage I (3.25) II (14.8%) III (52.7%) IV (29.4%) Tumor stage T1 (8.6%) T2 (23.6%) T3 (46.2%) T4 (21.5%) Node stage N0 (16.8%) N1 (36.6%) N2 (37.7%) N3 (9.0%)	M1 (12.4) M0 (87.6)	Smoking Chronic HBV infection Cardiovascular disease Diabetes mellitus Family history of NPC	46 (39–55)
Jung et al. [55]	2015	NLR PLR	119	Esophagus (100%)	Korea	retrospective study	male (94.1%) female (5.9%)	Pathological stage I (31.1%) II 33 (27.7%) III 49 (41.2)	M0 (100%)	NA	63.64 ± 8.42
He et al. [56]	2016	NLR PLR	317	Esophagus (100%)	China	retrospective study	male (84.5%) female (15.5%)	TNM I–II (68.5%) III–IV (31.5%)	M0 (53.9%) M1 (46.1%)	Smoking Alcohol consumption	60 (37–77)
Jiang et al. [57]	2016	NLR PLR	70	Thyroid (100%)	China	retrospective study	male (40%) female (60%)	Stage III or IV (50.0%)	M1 (2.9%)	NA	47.7 ± 13.9
Hirahara et al. [58]	2017	LMR NLR PLR	147	Esophagus (100%)	Japan	retrospective study	male (89.8%) female (10.2%	Pathological stage Ia–1b (40.1%) 2a–2b (22.4%) 3a–3c (37.4%)	NA	NA	NA
Ong et al. [59]	2016	LMR NLR PLR	133	Tongue (100%)	China	retrospective study	male (53.4%) female (46.6%)	pT classification, *n* (%) T1 (39.1%) T2 (60.9%)	M0 (96.2%) M1 (3.8%)	NA	51.92 (24–74)
Mao et al. [60]	2017	PLR	899	Larynx (100%)	China	retrospective study	male (97.1%) female (2.9%)	T 1 (22.9%) 2 (28.9%) 3 (29.1%) 4 (19.0%) N 0 (81.2%) 1 (9.7%) 2 (8.6%) 3 (0.6%)	M0 (100%)	Smoking Alcohol consumption	60 (22–87)
Dutta et al. [61]	2011	NLR PLR	112	Esophagus (100%)	UK	retrospective study	male (75.9%) female (24.1%)	TNM stage I 17.9% II 34.8% III 46.4% IV 0.9%	NA	NA	<65 (60.7%) 65–74 (33.9%) ≥75 (5.4%)
Li et al. [62]	2016	NLR PLR	409	Nasopharynx (100%)	China	retrospective study	male (70.4%) female (29.6%)	(I–II) (18.8%) (III–IV) (81.2%)	M0 (84.4%) M1 (15.6%)	NA	45 (18–77)
Messager et al. [63]	2015	PLR	153	Esophagus (100%)	UK	live patient samples	male (83.7%) female (16.3%)				
Hirahara et al. [64]	2016	LMR NLR PLR	147	Esophagus (100%)	Japan	retrospective study	male (89.8%) female (10.2%)	Pathological stage Ia–Ib 40.1% IIa–Iib 22.5% IIIa–IIIc 37.4%	NA	NA	NA
Moon et al. [65]	2015	NLR PLR	153	Oropharynx 33.3% Nasopharynx 31.4% Larynx 18.3% Hypopharynx 17.0%	Korea	live patient samples	male (84.3%) female (15.7%)	Clinical TNM stage T1/T2 17.6%/36.6% T3/T4 14.4%/31.4% N0/N1 21.6%/20.9% N2/N3 49.7%/7.8% Overall I/II 4.6%/17.0% Overall III/IV 13.7%/64.75%	M0 (100%)	Smoking Alcohol consumption	57 (16–78)
Hirahara et al. [66]	2016	LMR NLR PLR	147	Esophagus (100%)	Japan	retrospective study	male (89.8%) female (10.2%)	pathological stage 1a–1b (40.2%) 2a–2b (22.4%) 3a–3c (37.4%)	NA	NA	
Turri–Zanoni et al. [67]	2016	NLR PLR	215	Paranasal sinus (100%)	Italy	retrospective study	male (34%) female (66%)	pT classification pT1 19% pT2 18% pT3 22% pT4a 16% pT4b 25%	M0 (100%)	NA	65 (8–87)
Xie et al. [68]	2014	NLR PLR	317	Esophagus (100%)	China	retrospective study	male (77%) female (23%)	Tumor stage Stage I 88.4 Stage II 69.7 Stage III 42.4	M0 (100%)	NA	58.1 ± 8.9
Bojaxhiu et al. [69]	2018	NLR PLR	186	Oral cavity (28%) Oropharynx (45%) Hypopharynx (15%) Larynx (13%)	Switzerland	retrospective study	male (79%) female (22%)	UICC stage, N (%) I (3%) II (6%) III (24%) IV (68%)	M0 (100%)	Smoking Alcohol consumption	61 (41–88)
Sun et al. [70]	2017	NLR PLR	148	Nasopharynx (100%)	China	retrospective study	male (83.8%) female (16.2%)		M0 (100%)	Smoking	45 (24–72)
Sun et al. [71]	2015	NLR PLR	251	Nasopharynx (100%)	China	retrospective study	male (71.7%) female (28.3%)	UICC/AJCC stage I 2.4% II 15.9% III 47.4% IV 34.3%	M0 (100%)	NA	46 (15–76)
Tangthongkum et al. [72]	2017	PLR	274	Oral (100%)	Thailand	retrospective study	males (64.4%) females (35.6%)	NA	NA	NA	60 (21–92)
Ozturk et al. [73]	2016	NLR PLR	57	Tongue (100%)	Turkey	retrospective study	male (38.6%) female (61.4%)	stage I (64.9 %) stage II (35.1 %)	M0 (100%)	NA	57.8 (23–88)
Toyokawa et al. [74]	2016	NLR PLR	185	Esophagus (100%)	Japan	retrospective study	male (82.2 %) female (17.8%)	Clinical TNM stage I 36.2% II 42.2% III/IV 21.6%	M0 (100%)	NA	<65 51.4% ≥65 48.6%
Urabe et al. [75]	2017	LMR NLR PLR	1363	Resectable Gastric and Esophagogastric Junction	Japan	retrospective study	male (71.5%) female (28.5%)	T stage T1 58% T2 11.8% T3 17.6% T4 12.6%	M0 (100%)	NA	NA
Wang et al. [76]	2014	PLR	252	Upper aerodigestive tract	China	retrospective study	male (68.7%) female (31.3%)	Ann Arbor stage IE 61.5% IIE 38.5 %	NA	NA	41 (9–80)
Wei et al. [77]	2015	NLR PLR	423	Esophagus (100%)	China	retrospective study	male (80.6%) female (19.4)	TNM stage (AJCC, 7th) I (12.8%) II (39.7%) III (33.6%) IV (13.9%)	M0 (86.1) M1 (13.9)	NA	58 (24–88)
Xu et al. [78]	2015	NLR PLR	468	Esophagus (100%)	China	retrospective study	male (88.9%) female (11.1%)	Clinical stage I II IIIA IIIB + IIIC	NA	Smoking Alcohol consumption	58
Yang et al. [79]	2018	NLR PLR	515	Esophagus (100%)	China	retrospective study	male (81.2%) female (18.8%)	TNM Stage I II III	M0 (100%)	NA	61(33–92)
Ye et al. [80]	2018	NLR PLR	427	Nasopharynx (100%)	China	retrospective study	male (71.9%) female (28.1%)	TNM stage I 2.1 II 18.7% III 48.7% IV 30.5%	yes	NA	48 (17–82)
Yuan et al. [81]	2014	NLR PLR	327	Esophagogastric junction (100%)	China	retrospective study	male (86.2%) female(13.8%)	pTNM stage I and II (45.9%) III and IV (54.1%)	yes	NA	63.1± 9.7 (39–77)
Zhang et al. [82]	2017	NLR PLR	355	Esophagogastric junction (100%)	China	retrospective study	male (79.2%) female (20.8%)	TNM stage I, II (43.4%) III, IV (56.6%)	NA	NA	64 (34–82)
Chen et al. [20]	2018	NLR PLR MLR	361	Larynx (100%)	China	retrospective study	male (97.8%) female (2.2%)	TNM stage I 31.6% II 32.7% III 19.7% IV 16.0%	yes	NA	60 (35–87)
Hsu et al. [83]	2009	MLR	1069	Esophagus (100%)	Taiwan	retrospective study	male (94.5%) female (5.5%)	stage I 53 (10.9%) II 197 (40.4%) III 138 (28.3%) IV 100 (20.5%)	yes	NA	63.8 (34–88)
Chien et al. [84]	2016	MLR	2025	Esophagus (100%)	Taiwan	retrospective study	male (94.1%) female (5.9%)	cStage T1–2N0 T3–4N0 T1–2N (+) T3–4N (+) Unknown	yes	NA	55.2 ± 9.8
Furukawa et al. [85]	2019	LMR	103	Tongue (100%)	Japan	retrospective study	male (53.8%) female (46.2%)	stage I, II 84.5% III, IV 15.5%	yes	Smoking Alcohol consumption	63 (26–92)
Hsueh et al. [86]	2017	NLR PLR LMR	979	Larynx (100%)	China	retrospective study	male (97.5%) female (2.5%)	stage I 23.7% II 36.4% III 26.8% IV 13.1%	yes	NA	60.81 ± 9.68
Huang et al. [87]	2015	LMR	348	Esophagus (100%)	China	retrospective study	male (87.1%) female (12.9%)	NA	yes	NA	59.2 ±7.8
Kano et al. [88]	2016	NLR PLR LMR	285	Larynx (23.5%) Oropharynx (40.7%) Hypopharynx (35.8%)	Japan	retrospective study	male (88.4%) female (11.6%)	Clinical stage I, II (22.1%) III, IV (77.9%)	no	NA	61 (37–80)
Li et al. [89]	2013	LMR	1547	Nasopharynx (100%)	China	retrospective study	male 72.7% female 27.3%	Overall stage I-II (21.6%) III-IV (78.4%)	yes	NA	51 (6–87)
Li et al. [52]	2017	NLR PLR LMR	249	Nasopharynx (100%)	China	prospective study	male (73.9%) female (26.1%)	Clinical stage I-II 26.1 III-IV 73.9	yes	NA	≤50 (65.9%) >50 (34.1%)
Liu et al. [90]	2015	NLR PLR LMR	326	Esophagus (100%)	China	retrospective study	male (86.8%) female (13.2%)	T stage T1 (18.1%) T2 (18.4%) T3 (53.7%) T4 (9.8%)	NA	NA	59.2 ± 7.9 (38–80)
Oya et al. [91]	2018	NLR PLR LMR	441	Oral cavity 44% Larynx 28% Oropharynx 10% Hypopharynx 13 % Other 5%	Japan	retrospective study	male (73%) female (27%)	Stage I 32% II 18% III 15% IV 35%	no	NA	68 (27–92)
Yang et al. [92]	2018	NLR LMR	197	Hypopharyngeal (100%)	China	retrospective study	male (99.0%) female (0.01%)	Clinical stage I (2.0%) II (13.2%) III (27.5%) IV (57.4%)	yes	Smoking Drinking	<59 (50.8%) ≥59 (49.2%)

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
