# Peer review of "Prognostic Utility of Platelet–Lymphocyte Ratio, Neutrophil–Lymphocyte Ratio and Monocyte–Lymphocyte Ratio in Head and Neck Cancers: A Detailed PRISMA Compliant Systematic Review and Meta-Analysis"

_cancers, 2021, doi:10.3390/cancers13164166_

Round 1
Reviewer 1 Report
The manuscript is well written. The study is well conducted and methods are well designed.
Please check for extra spacings in the text.
Author Response
Reviewer 1
The manuscript is well written. The study is well conducted and methods are well designed.
Please check for extra spacings in the text.
Response: The highlighting editing issues have been rectified.
Reviewer 2 Report
The Authors of the manuscript assess prognostic value of Platelet-Lymphocyte-Ratio, Neutrophil-Lymphocyte-Ratio and Monocyte-Lymphocyte-Ratio in head and neck cancers. They conducted detailed PRISMA compliant systematic review and meta-analysis as showed in the supplementary material. In the Introduction profound rationale for the study is precisely described. The Authors emphasize that inflammatory response of the organism may influence cancer prognosis. Lymphocytes may stimulate the clearance of malignant cells, but the monocytes enable tumor progression by promoting angiogenesis and immunosuppression. All three potential prognostic markers, PLR, NLR and MLR have not been compared to each other so far and especially with regards to HNC cancers. Search strategy for this review, study selection, inclusion and exclusion criteria are properly described. Statistical analysis is described sufficiently.
In spite of above mentioned strengths there are some flaws in the manuscript that should be corrected before publication:
1) In the title three different words are used for description of analysed markers (Ratio, Ratios, Ratos) - the last one is grammatically incorrect, the same word should be used for all markers' description (Ratio).
2) Some English words should be written correctly i.e. 'Antagonising' instead of 'Antogonising' and 'cancer' instead of 'camcer'
3) p-values for all analyses should be reported precisely not as 'p<0.005'
4) In the paragraph titled 'Meta-analysis NLR subgroup' line 3 and line 6: the Authors describe levels of PLR whereas the paragraph deals with NLR - it should be corrected appropriately
5) In the paragraph titled 'Meta-analysis MLR subgroup' line 4 - the Authors describe level of PLR whereas the paragraph deals with MLR - it should be corrected appropriately
6) In the paragraph titled 'Meta-analysis MLR subgroup' the Authors stated that the overall pooled effect estimate (HR) for MLR had been found to be statistically significant, with a value of 1.002 (95% CI 0.720 – 1.396; p<0.05) - this seems to be incorrect and contradict basic statistical rules as hazard ratio with 95% confidence interval that contains '0' cannot be statistically significant. Figure 4 with forrest plot for MLR clearly shows that there is not any statistical significance observed. That is why I suggested in point 3 above reporting precise values of p. The Authors should double check their results and correct the manuscript appropriately.
7) The discussion is focused mainly on PLR and NLR, the Authors should expand on MLR as well.
8) In conclusion the Authors suggest that all three analysed markers have potential as prognostic markers but in fact the value of MLR seems to be not confirmed in the results.
Author Response
Reviewer 2
The Authors of the manuscript assess prognostic value of Platelet-Lymphocyte-Ratio, Neutrophil-Lymphocyte-Ratio and Monocyte-Lymphocyte-Ratio in head and neck cancers. They conducted detailed PRISMA compliant systematic review and meta-analysis as showed in the supplementary material. In the Introduction profound rationale for the study is precisely described. The Authors emphasize that inflammatory response of the organism may influence cancer prognosis. Lymphocytes may stimulate the clearance of malignant cells, but the monocytes enable tumor progression by promoting angiogenesis and immunosuppression. All three potential prognostic markers, PLR, NLR and MLR have not been compared to each other so far and especially with regards to HNC cancers. Search strategy for this review, study selection, inclusion and exclusion criteria are properly described. Statistical analysis is described sufficiently.
In spite of above-mentioned strengths there are some flaws in the manuscript that should be corrected before publication:
1) In the title three different words are used for description of analysed markers (Ratio, Ratios, Ratos) - the last one is grammatically incorrect, the same word should be used for all markers' description (Ratio).
Response: The title has been changed according to the recommendation and consistency with the word ‘Ratio’ has been maintained.
2) Some English words should be written correctly i.e. 'Antagonising' instead of 'Antogonising' and 'cancer' instead of 'camcer'
Response: The indicated spelling issues have been corrected.
3) p-values for all analyses should be reported precisely not as 'p<0.005'
Response: Thank you for the suggestion. Changes have been made to indicate the exact p-values for all statistical analyses.
4) In the paragraph titled 'Meta-analysis NLR subgroup' line 3 and line 6: the Authors describe levels of PLR whereas the paragraph deals with NLR - it should be corrected appropriately
Response: Indicated corrections have been made.
5) In the paragraph titled 'Meta-analysis MLR subgroup' line 4 - the Authors describe level of PLR whereas the paragraph deals with MLR - it should be corrected appropriately
Response: Indicated corrections have been made.
6) In the paragraph titled 'Meta-analysis MLR subgroup' the Authors stated that the overall pooled effect estimate (HR) for MLR had been found to be statistically significant, with a value of 1.002 (95% CI 0.720 – 1.396; p<0.05) - this seems to be incorrect and contradict basic statistical rules as hazard ratio with 95% confidence interval that contains '0' cannot be statistically significant. Figure 4 with forrest plot for MLR clearly shows that there is not any statistical significance observed. That is why I suggested in point 3 above reporting precise values of p. The Authors should double check their results and correct the manuscript appropriately.
Response: Thank you for bringing this error to notice. Your recommendations have been followed up and the incorrect result has been changed accordingly to indicate a non-significant result for the MLR subgroup.
7) The discussion is focused mainly on PLR and NLR, the Authors should expand on MLR as well.
Response: Thank you for the suggestion. The discussion has been expanded to include insights on MLR as a prognostic indicator in HNC.
8) In conclusion the Authors suggest that all three analysed markers have potential as prognostic markers but in fact the value of MLR seems to be not confirmed in the results.
Response: As per the issue highlighted, the error regarding MLR has been rectified, with changes made to the conclusion.
Reviewer 3 Report
In this manuscript, the authors reviewed and analyzed the prognostic efficacy of three prognostic markers PLR, NLR and MLR and the prognostic potential of these three markers in HNC in 49 studies. The results showed that PLR, NLR and MLR were significantly correlated with poorer OS, which suggested that PLR, NLR and MLR ratios could be powerful prognostic markers in Head and Neck Cancers that could guide treatment. It’s great that the authors provided the details of the methods, however, the conclusion and the discussion were too short. The authors should add the discussion to complete this manuscript and help the readers to understand the results and the meaning of this study.
Author Response
Reviewer 3
In this manuscript, the authors reviewed and analyzed the prognostic efficacy of three prognostic markers PLR, NLR and MLR and the prognostic potential of these three markers in HNC in 49 studies. The results showed that PLR, NLR and MLR were significantly correlated with poorer OS, which suggested that PLR, NLR and MLR ratios could be powerful prognostic markers in Head and Neck Cancers that could guide treatment. It’s great that the authors provided the details of the methods, however, the conclusion and the discussion were too short. The authors should add the discussion to complete this manuscript and help the readers to understand the results and the meaning of this study.
Response: Thank you for the suggestion. The discussion and the conclusion have been expanded upon with insights regarding MLR as a prognostic marker, and the utility of this study and where it currently stands in informing regarding the utility of PLR, NLR and MLR as prognostic markers.
We would like to thank all the reviewers for their time and effort in reviewing the study and providing feedback, so that we may improve the quality of our study.